Empirical mode decomposition using deep learning model for financial market forecasting

Jin Zebin zebin.jin.quc@gmail.com 1
Jin Yixiao 2
Chen Zhiyun 3
1 College of Management, Ocean University of China , Qingdao , Shandong , China
2 Shanghai Yingcai Information Technology Ltd. , Fengxian , Shanghai , China
3 Jinan University , Nanshan , Shenzhen , China
Alarcon-Aquino Vicente
Electronic publication date: 2022 Sep 14
Publication date: 2022
Volume: 8
Electronic Location ID: e1076
Received 2022 Jun 9; Accepted 2022 Aug 8
Copyright: ©2022 Jin et al.
Copyright year: 2022
Copyright holder: Jin et al.
License: This is an open access article distributed under the terms of the Creative Commons Attribution License, which permits unrestricted use, distribution, reproduction and adaptation in any medium and for any purpose provided that it is properly attributed. For attribution, the original author(s), title, publication source (PeerJ Computer Science) and either DOI or URL of the article must be cited.
License URL: https://creativecommons.org/licenses/by/4.0/

Keywords: Deep learning, Decision making and analysis, EMD, Eigenmode function, Interval EMD, Particle swarm optimization, Time series

Funding: The authors received no funding for this work.

==============================
Financial market forecasting is an essential component of financial systems; however, predicting financial market trends is a challenging job due to noisy and non-stationary information. Deep learning is renowned for bringing out excellent abstract features from the huge volume of raw data without depending on prior knowledge, which is potentially fascinating in forecasting financial transactions. This article aims to propose a deep learning model that autonomously mines the statistical rules of data and guides the financial market transactions based on empirical mode decomposition (EMD) with back-propagation neural networks (BPNN). Through the characteristic time scale of data, the intrinsic wave pattern was obtained and then decomposed. Financial market transaction data were analyzed, optimized using PSO, and predicted. Combining the nonlinear and non-stationary financial time series can improve prediction accuracy. The predictive model of deep learning, based on the analysis of the massive financial trading data, can forecast the future trend of financial market price, forming a trading signal when particular confidence is satisfied. The empirical results show that the EMD-based deep learning model has an excellent predicting performance.

Introduction

Due to the huge volume of information, extracting a meaningful piece of information becomes a difficult task. Deep learning models are considered the best information extractors and classifiers for financial market trend forecasting by using a huge volume of dynamic information. Recent research on deep learning applications for financial market trend predictions illustrates that long and short-term memory neural networks, convolutional neural networks and their combination forms are regularly used in deep learning (Dias et al., 2020; Hu, Zhao & Khushi, 2021; Nosratabadi et al., 2020; Ozbayoglu, Gudelek & Sezer, 2020).

Financial market trend forecasting become an important topic and has attracted constant attention in finance (Haq et al., 2021; Jushi et al., 2021; Migliorelli, 2021; Umar et al., 2021). Nowadays, it is extensively used in different companies of various disciplines to predict financial markets, which makes market forecasting a promising financial research topic (Buczynski, Cuzzolin & Sahakian, 2021; Rouf et al., 2021).

Financial data forecasting by analysing huge amounts of raw data has always been a vital issue in the economic domain (Jan, 2021). Existing forecasting methods exhibit few symptoms of discomfort in several stages of analysis (Sivarajah et al., 2017). Conventional artificial intelligence with the non-linear feature is not able to model complex data accurately yet, which contains traditional methodology, equations, parameters, and high dimensional noisy time series financial sequences (Di Franco & Santurro, 2021; Hijazi, Al-Dahidi & Altarazi, 2020; Längkvist, Karlsson & Loutfi, 2014).

Over the last few years, empirical mode decomposition (EMD) has been considered one of the efficient approaches for the improvement of financial market forecasting (Huang et al., 1998; Nava, Matteo & Aste, 2018). The EMD approach decomposes the original signal within a finite set of approximately orthogonal oscillating elements, they are called intrinsic mode functions (IMFs) (Souza, Escola & Brito, 2022). IMFs have particular time scales of oscillations determined by the own maximum and minimum of the data, which are retrieved by the information itself without depending on any other function.

Most of the previous works only analyzed the closing market price (Jin, Yang & Liu, 2020; Li & Wu, 2021). The financial time series (FTS) is a unique interval time series. The stock index fluctuates between the highest price and the lowest price every day (Ananthi & Vijayakumar, 2021; Zhang & Lou, 2021). If only the closing market price is considered, much useful information will be lost. Therefore, the interval EMD algorithm is introduced for detecting the closing market price, the highest price, and the lowest price of each index. However, when detecting the highest and lowest prices, it shows a better effect. These results show that the interval EMD algorithm performs better in detecting the highest and lowest prices in the FTS intervals.

In this research, the interval EMD algorithm is used with BPNN for the particular structure of FTS, which includes time, opening market price, the highest price, the lowest price, closing market price, and transaction volume. Financial market transaction data are analyzed and optimized using PSO, so that the time series can be understood from different frequency scales, thereby revealing the intrinsic laws of data. The empirical demonstrated that the EMD-based deep learning model has an excellent predicting performance. Moreover, in all aspects of statistical comparison, our proposed forecasting method performed better than the benchmark method.

The rest of this article is organized as follows. In Section 2, the recent literature review is discussed to find out the research gap. Section 3 contains the methodology, which describes the model and experimental flow. Section 4 provides the detailed finding of this research as results and discussion. Finally, Section 5 draws the conclusion and suggestions for future work.

Literature Review

In the previous research, there are relatively a small number of studies found that analysis and compare the time series of financial data in high frequency to low-frequency IMFs. Asset prices evaluation is determined by various factors mostly timescales and short-term to long-term price fluctuations (Ahmed, 2022; Chhajer, Shah & Kshirsagar, 2022; Urom et al., 2021). Several market surveys and empirical research suggest that numerous financial time series mostly exhibit nonstationary characteristics, such as time-depending volatility and market trends (In & Kim, 2012; Leung & Zhao, 2021; Maghyereh, Awartani & Abdoh, 2019; Yahya, Oglend & Dahl, 2019; Yu, 2019). The articles (Li et al., 2021; Yu, Wang & Lai, 2008) demonstrated that timescale decomposition is an actual efficient method that followed the “divide-and-conquer” approach. For example, the divide-and-conquer approach has been used in several fields: oil prices (Rădulescu et al., 2020; Wang et al., 2018) foreign currency exchange rate (Jin et al., 2021; Lin, Chiu & Lin, 2012; Wang & Luo, 2021), stock market trend (Cheng & Wei, 2014; Na & Kim, 2021; Stasiak, 2020; Wang & Luo, 2021), wind speed (Hu et al., 2021; Wang et al., 2014; Xie et al., 2021), electronics sales (Chen & Lu, 2021; Lu & Shao, 2012), healthcare (Aileni, Rodica & Valderrama, 2016; Dwivedi et al., 2019; Singh, Dwivedi & Srivastava, 2020), and tourism market (Chen, Lai & Yeh, 2012; Guerra-Montenegro et al., 2021; Tang et al., 2021). The hybrid EMD combined with the artificial neural network(ANN) method was applied to predict the first, second, and third steps moving forward wind speed time series (Chen et al., 2021; Hu et al., 2021; Liu et al., 2012; Liu, Hara & Kita, 2021). Several predicting powers from low to high frequency and short-term to long-term trend elements were observed for analysis of the accuracy of EMD forecasting combined with ANN in the Baltic Exchange Dry Index (Gavriilidis et al., 2021; Zeng & Qu, 2014). Based on the literature, EMD is mostly executed on the whole dataset before forecasting, which means that future forecasted data are utilized to develop the EMD (Buczynski, Cuzzolin & Sahakian, 2021; Chen, Lai & Yeh, 2012; Liu, Mi & Li, 2018; Lu & Shao, 2012; Na & Kim, 2021).

Long, Lu & Cui (2019), demonstrated an end-to-end model named multi-filters neural network for knowledge mining on financial time-series and price fluctuation data. Hierarchical keyword-based attention networks as the HKAN model described in Wu et al. (2019), analyse the trading trend, and stock messages. LSTM, Seq2seq, and Wavenet methods were used in the article (Cho et al., 2019) to forecast the stock price. Financial news and sentiment dictionary-based model presented in Lien Minh et al. (2018) to predict stock prices strand. To avoid the expensive annotation article (Hu et al., 2018) proposed a candlestick charts-based method with a synthesis technique to present price history for price forecasting. Based on the recent sequence of related news and self-paced learning a model is described in the article (Hu et al., 2018) to forecast stock market trends. Article (Kim & Khushi, 2020), 2D gated transformer method described which refer reinforcement learning and agent incorporating to forecast market trend. Genetic algorithm with the crossover technique used in the article (Zhang & Khushi, 2020) for forecasting the financial market which overcomes traditional trading strategy limitations. Several deep learning methods were used in the article (Shi et al., 2019) to perform better analysis and design to forecast market trends. None of the models in Table 1, have used PSO for parameter optimization which may increase the forecasting accuracy to a certain level.

Table 1 Summary of recent research for market trend forecasting using deep learning.

Ref.	Variable	Dataset	Model	
Long, Lu & Cui (2019)	Open price, close price, low price, high price, and transaction volume	CSI 300 Index	Multi-manifold feature fusion	
Wu et al. (2019)	Market news messages such as title, keywords, and summary	FTSE 100	HAN	
Cho et al. (2019)	Open price, close price, low price, high price, and transaction volume, MACD, CCI, ATR, BOLL, MA5, MOM6, ROC, RSI, exchange rate, WVAD, and interest rate	CTBC HOLDINGS, ESFH, Fubon Financial and FFHC	Wavenet	
Lien Minh et al. (2018)	Open price, close price, low price, high price, and transaction volume, and stochastic oscillator	The Standard and Poor’s 500, and Vietnam Ho Chi Minh Stock Index,	Stock2Vec Embedding BGRU	
Hu et al. (2018)	Candlestick charts	FTSE 100	Convolutional AutoEncoder	
Hu et al. (2018)	Close price, transaction volume, and news sequence	Chinese stock price but not given any specific data	HAN	
Kim & Khushi (2020)	Open price, close price, low price, high price, and transaction volume	MMM stock, JPM, PG, AAPL, UNH, WMT, XOM, DD, and VZ	2D Gated Transformer	
Zhang & Khushi (2020)	Moving average(MA), Exponential MA, double exponential MA, triple exponential MA, and relative strength index.	Forex exchange rates data	Genetic Algorithm	
Shi et al. (2019)	News and financial data	Apple Inc. and SPX	CNN, LSTM, and Hybrid of RNN	

Over time, institutional investors preferred adopting financial econometric models to analyse financial data and study market features (Buturac, 2021; Datta et al., 2021; Messeni Petruzzelli, Murgia & Parmentola, 2021). The analysis results of the econometric model are often explanatory. However, as the amount of financial transaction data increases sharply, the form of data is increasingly diversified (including structured data of trading quotes, and unstructured data such as financial news), which makes transactions complicated. Consequently, diversified data forms make it increasingly challenging to model mathematical equations entirely. The deep learning method provides a new idea; it finds the laws of big data, enables the model to autonomously mine the statistical laws hidden behind the data, and guides the financial transactions (Domingos, Ojeme & Daramola, 2021; Park et al., 2021; Shukla, Muhuri & Abraham, 2020).

For the forecasts of these macroeconomy and financial markets, there are many applications of the EMD-based separation and integration DNN model (Huang et al., 2018; Kyriazis, 2021; Nguse et al., 2021; Petropoulos et al., 2021; Zhang, Nakajima & Hamori, 2021). Most scholars have a particular preference for EMD. However, mostly, previous explorations employed EMD for general univariate or multivariate time series rather than a unique structure for FTS. Therefore, a unique EMD is proposed for the particular structure of FTS, which includes time, opening market price, the highest price, the lowest price, closing market price, and transaction volume. Notably, the interval EMD algorithm for FTS utilizes the highest price for calculating the upper envelope and the lowest price for the lower envelope. In this way, the IMF obtained can better demonstrate the magnitude of local shocks. This study used the EMD method, instead of other signal decompositions as EMD makes more sense since financial time series are non-stationary and it can extract the main trend of these signals and suppress noise.

Methodology

The key of this method is empirical mode decomposition, which can decompose complex signals into a finite number of IMFs. The decomposed IMF components contain local characteristic signals of different time scales of the original signal (Nan et al., 2018). The EMD method can make the non-stationary data smoothing, and then perform the Hilbert transform to obtain the time spectrum map to obtain the frequency with physical meaning (Fu, 2018). Compared with short-time Fourier transform, wavelet decomposition and other methods, this method is intuitive, direct, posteriori and adaptive, because the basic function is decomposed by the data itself. Since the decomposition is based on the local characteristics of the signal sequence time scale, so it is considered adaptive.

In this research, multilayer perceptrons (MLPs) were used including the feedforward and BP method. The feedforward and BP method were used to increase the MLPs accuracy which is very important for financial market forecasting. This type of NN falls under the supervised networks therefore they require accurate output for learning. The architecture consists of three layers named the input, hidden and output layers. In this study, a standard 64 neurons are used in the hidden layer. In reality, this architecture shows the great approximate performance of optimal statistical classifiers in complex problems. MLPs architecture was chosen for the nature of this experiment and this is the most common network architecture used for financial market forecasting.

Figure 1 shows the proposed approach for financial market forecasts. It takes huge amounts of transactional data and performs feature extraction. After that used feature extraction in the prediction model to do the market forecasts. The individual IMF forecasts are made on actual input data in a single-day way within the sliding window. The hyperparameters for the BPNN of the proposed model are selected back-to-back concerning the associated reduction in out-of-sample loss. Each intrinsic function is forecast with a BPNN. Then, all the predicted individual components are combined to obtain the overall predicted signal.

Figure 1 Flowchart of the proposed (EMD + BPNN) approach for financial market forecasts.

Deep learning model based on EMD

EMD is a spectrum analysis method proposed by NASA’s signal processing expert (Huang et al., 1998), which analyzes nonlinear and non-stationary data series. EMD is also known as Hilbert-Huang Transform (HHT). It includes two processes: EMD and HHT. Any complex signal can be decomposed into several IMFs through EMD, and the number of IMFs is often limited. These IMF series can well describe each local oscillation of the original data series, with well-performed HHT. Therefore, the Hilbert spectrum obtained has excellent energy time-frequency features (Fu, 2018; Nan et al., 2018).

Algorithm 1: EMD	
Input: A signal St	
i). Set rt:=St and k = 0	
Whilert is not distinct do	
ii). Set mt=rt	
Whilemt!= 0 do	
iii). Interpolate between min (respective max), ending up with some ‘envelope’ emintrespectiveemaxt.	
iv) Calculate the average mt=emint+emaxt2	
v) Extract ct=rt−mt,andsymbolizedctasrt	
end while	
vi). Set k = k + 1	
vii). Set IMFkt=ct	
viii). Set rt=St−∑k=1KIMFkt	
end while	
output St=∑k=1KIMFkt	

The EMD method considers that any signal is composed of several eigenmode functions. A signal can contain thousands of eigenmode functions at any time. If the eigenmode functions overlap each other, a composite signal is formed. EMD decomposition aims to obtain the eigenmode function and then perform HHT on each eigenmode function, thereby obtaining the Hilbert spectrum. In this case, the original signal is mentioned below. (1) St= ∑k=1KIMFkt+rKt.

In Eq. (1), S(t) express the original signal which iteratively decomposes a time series, IMFkt is are called intrinsic mode functions,plus a nonoscillatory trend called the residual term expressed by rKt and K-level IMF is obtained after EMD decomposition (Zhu et al., 2019), where k = 1, 2, …, K. (2) rKt=St−∑k=1KIMFkt.

By rearranging Eq. (1), the residual term can be calculated by Eq. (2). Individually, IMFs at all levels are not acquired by explicit convolution calculations; instead, they are obtained through an algorithm. After IMFs at all levels are obtained through the algorithm, each IMF can be a decomposition series and substituted into the BPNN of separation and integration. Therefore, a complete EMD-based separation and integration BPNN model is built. Here, the concept of level is generated by multiple EMD iterations of the data. Therefore, it does not correspond to a strict level of time scale. It is a scale-space representation that reflects the features of local oscillations (Fang et al., 2018; Luo et al., 2019; Ullah et al., 2018; Wen, Gao & Li, 2019). In general, the choice of the objective function is determined by the specific problem. If it is a backpack problem, fitness is the total price of the object in the package (Zhang, Han & Deng, 2018).

FTSEMD

FTS contains information that is different from the general time series in contents and formats. Therefore, financial time series empirical mode decomposition (FTSEMD) is also different from the general time series. Generally, an FTS can be represented by five-time series as Eq. (9). (3) Xt=X.Ot,X.Ht,X.Lt,X.Ct,X.Vt.

In Eq. (3), X.Ot is the time series of opening market price, X.Ht is the time series of the highest price, X.Lt is the time series of the lowest price, X.Ct is the time series of closing market price, and X.Vt is the time series of transaction volume. The FTSEMD can utilize various combinations of the above times series to perform EMD. Precisely, the FTS exhibits nonlinear, non-stationary, multiscale, and interval features, its EMD processing is also different from that of the general time series. The daily price of FTS fluctuates between the highest price and the lowest price. Therefore, when building a mathematical model to predict its fluctuation trend, all the information on transaction prices must be thoroughly considered. The conclusions drawn by modelling only with the closing market price will be biased because it ignores other transaction prices. The aim is to build a unique structure of FTS. Hence, an interval EMD algorithm is proposed, which combines the time series of the highest price, the lowest price, and the forecast signal.

Dimensionality reduction after FTSEMD

During processing, multiple regression models often contain more explanatory variables; besides, these variables are correlated, and the information they contained overlaps, making the analysis more complicated. Therefore, for the solutions to the above problems and demonstration of the essential features of the original data, variables that are connected are often indicated by several indicators. This process is dimensionality reduction. Afterwards, these indicators, which will be unconnected, contain most of the information in the original data, thereby benefiting the mathematical modelling.

The second crucial step of the FEPA model is reducing the dimensionality of the IMF components after EMD. The FTS extract data through the forward-scrolling window, and many IMF components are obtained through EMD. Due to the scrolling feature, most of the data entered into the scrolling window are the same each time, except that the last batch of data is deleted, and the latest batch of data is added. After EMD, the data, which are extracted by the forward-scrolling window, contains much redundant information; hence, dimensionality reduction is necessary. Here, the PCA algorithm is adopted to reduce the dimensions of the decomposed IMF components. Afterwards, several principal components, containing most of the information of the original signals, are obtained. The cumulative variance contribution rate of these components must meet particular conditions. The PCA dimensionality reduction after FTSEMD is a significant innovation, and PCA is an essential step in FEPA modelling.

PSO

The PSO algorithm was first presented in Kennedy & Eberhart (1995). The PSO is a random search strategy based on a population of particles. The principle concept of PSO reaches from the social behaviour of flocks birds. In this algorithm, each particle drives in a D-dimension based on its own experience and other particles as well. The PSO algorithm is easy to understand, simple to code, and easy to implement. However, the setting of parameters has a great influence on the performance of the algorithm, such as control convergence, avoiding premature and so on Li et al. (2018); Liang et al. (2018); Wang & Li (2017); Wei-Chang et al. (2010). In PSO, the position of particle i can be represented by the D dimension vector in Eq. (4). (4) Xit=xi1,xi2,xi3,…,xiD.

The velocity at the time is expressed by Vit which is calculated using Eq. (5). (5) Vit=vi1,vi2,vi3,…,viD.

The best position of the particle itself is expressed by Pit which is calculated using Eq. (6). (6) Pit=Pi1,Pi2,Pi3,…,PiD.

The current optimal position of the entire particle swarm is expressed by Pgt which is calculated using Eq. (7). (7) Pgt=Pg1,Pg2,Pg3,…,PgD.

The tth generation particle updates velocity and position expressed by Vit and Xitwhich are calculated using Eqs. (8) and (9) (Liu et al., 2021).

(8) Vit=wVit−1+c1r1Pi−Xit−1+c2r2Pg−Xit−1

(9) Xit=Xit−1+Vit.

The EMD algorithm and PSO algorithm are used to extract the original FTS and obtain the dataset. Then, the dataset is decomposed into eigenmode functions with different scales by the EMD method; meanwhile, the PSO algorithm is used for parameter optimization and prediction. Combining the EMD and PSO algorithms can understand the data features from multiple dimensions, which will effectively improve the control over financial market transactions and accurately predict future financial market transactions.

Results and Discussion

To illustrate the proposed model, the daily exchange rates of four major currency pairs related to CNY from January 01, 2011 to May 31, 2021 in total 2716 days records are used as the experimental dataset. We used the first 2616 days records from January 01, 2011 to January 11, 2021 as training data to train the system. The four major currency pairs used are USD/CNY, EURO/CNY, JPY/CNY, and CHF/CNY used. Taking the Shanghai, Shenzhen, Hang Seng, and Dow Jones stock market index data as an example, we construct several data points on the aforementioned period within the length of slide windows which are selected as 10, 20, 30, 50, 60, 70, 80, 90, and 100, respectively. The latest 100 days records from January 12, 2021 to May 31, 2021 are used to compare the forecast results.

Empirical analysis of the prediction effect of interval EMD model

EMD decomposes IMFs successively through multiple screening processes, during which the local average of signals is calculated from their upper and lower envelopes of them. The upper and lower envelopes are the local maxima and minima of the signal given by the spline interpolation algorithm. Since both ends of the signal cannot be at the maximum and minimum values at the same time, the upper and lower envelopes will inevitably appear divergently at both ends of the data series. Errors are introduced into the screening process (Cai et al., 2017). As the screening process continues, the result of such divergence will gradually “contaminate” inward the entire data series, causing severe distortions in the results. For long data series, data at both ends can be discarded according to the extreme point, thereby ensuring that the resulting envelope distortion is minimized. However, for short data series, discarding data at both ends becomes completely infeasible.

In general, fluctuations in data series of trading prices in the financial market are random, nonlinear, and non-stationary. The current prediction model is difficult to fully understand the features of various types of data and obtain good prediction results. If a model has an excellent predictive ability for trading prices in the financial market, its value is self-evident.

As a new method to process nonlinear and non-stationary signals, EMD time-frequency analysis is fundamentally different from traditional signal time-frequency analysis methods and has achieved excellent results in practical applications. The EMD decomposition algorithm obtains the IMF components of the signal feature scales at different time points through layer-by-layer screening (Nait Aicha et al., 2018). The primary goal of EMD decomposition is to smooth the signal, perform HHT on the IMF component, and finally, obtain the instantaneous frequency component corresponding to the IMF component. The instantaneous frequency obtained has a reasonable physical meaning. The Hilbert spectrogram obtained is a two-variable function of time and frequency, from which the frequency information at any time can be obtained (Zhang & Zeng, 2017). For example, the magnitude and amplitude of the frequency, as well as the corresponding moments appearing, can be obtained, which can describe the time-frequency features of the non-stationary and nonlinear signal in detail. (10) MAE=1n×∑i=1nTi−Ai

(11) MAPE=1n×∑i=1nTi−AiTi×100%

(12) RSME=1n×∑i=1nTi−Ai2

(13) HitRate%=Correct PredictionsNumber of Test Data∗100

(14) DS=100n×∑u=1ndi

(15) di=1Ti−Ti−1Ai−Ai−1≥00otherwise

(16) MAD=∑|Ti−Ai|n

(17) TS=∑Ti−AiMAD.

Equations (10), (11), (12), (13), (16) and (17) are used to calculate the mean absolute error (MAE), mean absolute percentage error (MAPE), root mean squared error (RMSE), hit rate percentage, mean absolute deviation (MAD), and the tracking signal (TS), respectively where Ti and Ai express the actual and forecast value. Although obtaining a precise prediction of the stock index is challenging, a rough prediction of the price trend will help in investment decisions. The EMD algorithm decomposes the time series of the stock index and produces a stationary IMF series, which improves the predictive ability of the model. The time series can be mastered from different scales to reveal the intrinsic laws of data.

Empirical analysis of major financial markets

The financial market is undoubtedly complex, uncertain, and dynamic. In the financial markets, people cannot use a single strategy or model simultaneously; otherwise, they may suffer huge losses. The same model may behave differently in different financial environments. Has the model been applied since many years ago? Is the mature experience of other countries also applicable to the Chinese market? Answers to these questions are unknown. Nevertheless, Shanghai, Shenzhen, Hang Seng, and Dow Jones stock market index data can describe the problems in applications, and the above questions can be answered by empirical analysis.

At present, artificial intelligence (AI) has been widely used in the Internet and manufacturing industries (Hansen & Bøgh, 2021; Qiu, Suganthan & Amaratunga, 2017; Rizvi et al., 2021; Zeba et al., 2021). Whether for the continuous expansion of application fields or the continuous optimization of deep learning algorithms, AI has dramatically improved the traditional working and thinking modes. According to the information collected from news and reports, AI can comprehensively consider whether the content in the collected information is positive or negative for the fundamentals of the financial market; then, it rates the information as very bad, bad, moderate, good, and very good. The deep learning model is trained to determine whether an article conveys positive or negative information through fundamental logic (Zhu et al., 2018). AI algorithms show the excellent ability of market background interpretation in various stages of back-tests. Also, the risk preference index has individual rationality, which shows better early-warning capability when the market upswing terminates. If the market index increases but the risk preference index decreases, the market will be prompted to withdraw the risks, and the effect is noticeable. As the model learns information, the sentiment index and risk preference measures will become accurate.

In Table 2, the statistics of the Shenzhen Index Yield indicate that the kurtosis is 2.286, with a thick tail phenomenon. The thick tail characteristic of the negative deviation direction is more evident than the index. Therefore, the market of the Shenzhen Stock Exchange is more volatile and dynamic than that of the Shanghai Stock Exchange. However, Shanghai Stock Exchange is also more volatile and dynamic than Hang Seng and Dow Jones respectively based on kurtosis.

The EMD-with BPNN has a higher hit rate than other single reference models, and its prediction error is also small. Hence, the EMD algorithm can improve the prediction accuracy of neural networks. The above results indicate that principal component analysis (PCA) can reduce data dimensionality, compress redundant data, improve prediction accuracy, and shorten the data training time of neural networks.

Table 3 provided the evaluation of the forecast model based on RMSE, MAPE, MAE and TS. In all aspects of statistical comparison, our proposed forecasting method performed better than the benchmark RW method.

Figure 2 expresses the IMF1, IMF2, IMF3, and IMF4 component map on the provided data for the US dollar against the CNY exchange rate. The comparison of USD to CNY exchange rate’s actual data and forecast value as the output system is expressed in Fig. 3. The upper part of Fig. 3 represents the graphical view of actual data and forecast data for visualization of the differences. Moreover, the bottom part of Fig. 3 represents the curve fitting plots of USD to CNY exchange rate for the forecast versus actual data. In the case of USD to CNY exchange rate forecasting proposed method’s accuracy in terms of RMSE, MAPE, MAE, and TS are reported as 0.011061, 0.001423, 0.009247, and 10.36, respectively. Similarly, benchmark method RW accuracy in terms of RMSE, MAPE, MAE, and TS are reported as 0.037337, 0.004162, 0.027068, and −66.88, respectively. From this comparison, it is clear that the proposed method performs better than the benchmark method for USD to CNY exchange rate forecasting.

Table 2 Shanghai composite, Shenzhen, Hang Seng, and Dow Jones index yield statistics.

Index yield	N	Min.	Max.	Mean	SD	Skewness	Kurtosis	
Statistics	Statistics	Statistics	Statistics	Statistics	Statistics	Statistics	SD	Statistics	SD	
Shanghai Composite	8638	−0.0924	0.0922	0.0002	0.0188	−0.460	0.040	3.566	0.089	
Shenzhen	8638	−0.0945	0.0963	0.0003	0.0198	−0.401	0.040	2.286	0.096	
Hang Seng	8658	−0.1288	0.1437	0.0002	0.1680	0.301	0.046	8.687	0.096	
Dow Jones	8608	−0.0775	0.1108	0.0002	0.0141	0.136	0.046	10.169	0.100	

Table 3 Evaluating the forecast model.

Model	Accuracy factor	USD/CNY	EURO/CNY	JPY/CNY	CHF/CNY	
Proposed Forecast	RMSE	0.011061	0.018999	0.000206	0.019752	
MAPE	0.001423	0.002051	0.002450	0.002249	
MAE	0.009247	0.016009	0.000149	0.016032	
TS	10.36	−18.44	−0.32	−18.09	
RW Benchmark	RMSE	0.037337	0.039198	0.004327	0.165387	
MAPE	0.004162	0.004184	0.049092	0.0139802	
MAE	0.027068	0.032679	0.002951	0.100183	
TS	−66.88	−47.80	−72.49	−50.31	

Figure 2 IMF component map of the US dollar against the CNY exchange rate.

Figure 3 USD to CNY exchange rate forecast and actual graph (for interpretation of the references to colour in this figure legend).

Figure 4 expresses the IMF1, IMF2, IMF3, and IMF4 component map on the provided data for the EURO against the CNY exchange rate. The comparison of EURO to CNY exchange rate’s actual data and forecast value as the output system is expressed in Fig. 5. The upper part of Fig. 5 represents the graphical view of actual data and forecast data for visualization of the differences. Moreover, the bottom part of Fig. 5 represents the curve fitting plots of EURO to CNY exchange rate for the forecast versus actual data. For EURO to CNY exchange rate forecasting proposed method’s accuracy in terms of RMSE, MAPE, MAE, and TS are reported as 0.018999, 0.002051, 0.016009, and −18.44, respectively. Similarly, benchmark method RW accuracy in terms of RMSE, MAPE, MAE, and TS are reported as 0.039198, 0.004184, 0.032679, and −47.80, respectively. From this comparison, it is clear that the proposed method performs better than the benchmark method for EURO to CNY exchange rate forecasting. Figure 6 expresses the IMF1, IMF2, IMF3, and IMF4 component map on the provided data for the JPY against the CNY exchange rate. The comparison of JPY to CNY exchange rate’s actual data and forecast value as the output system is expressed in Fig. 7. The upper part of Fig. 7 represents the graphical view of actual data and forecast data for visualization of the differences. Moreover, the bottom part of Fig. 7 represents the curve fitting plots of JPY to CNY exchange rate for the forecast versus actual data. In the case of JPY to CNY exchange rate forecasting proposed method’s accuracy in terms of RMSE, MAPE, MAE, and TS are reported as 0.000206, 0.002450, 0.000149, and −0.32, respectively. Similarly, benchmark method RW accuracy in terms of RMSE, MAPE, MAE, and TS are reported as 0.004327, 0.049092, 0.002951, and −72.49, respectively. From this comparison, it is clear that the proposed method performs better than the benchmark method for JPY to CNY exchange rate forecasting.

Figure 4 IMF component map of the EURO against the CNY exchange rate.

Figure 5 EURO to CNY exchange rate forecast and actual graph (for interpretation of the references to colour in this figure legend).

Figure 6 IMF component map of the JPY dollar against the CNY exchange rate.

Figure 7 JPY to CNY exchange rate forecast and actual graph (for interpretation of the references to colour in this figure legend).

Figure 8 expresses the IMF1, IMF2, IMF3, and IMF4 component map on the provided data for the CHF against the CNY exchange rate. The comparison of CHF to CNY exchange rate’s actual data and forecast value as the output system is expressed in Fig. 9. The upper part of Fig. 9 represents the graphical view of actual data and forecast data for visualization of the differences. Moreover, the bottom part of Fig. 9 represents the curve fitting plots of CHF to CNY exchange rate for the forecast versus actual data. For CHF to CNY exchange rate forecasting proposed method’s accuracy in terms of RMSE, MAPE, MAE, and TS are reported as 0.019752, 0.002249, 0.016032, and −18.09, respectively. Similarly, benchmark method RW accuracy in terms of RMSE, MAPE, MAE, and TS are reported as 0.165387, 0.0139802, 0.100183, and −50.31, respectively. From this comparison, it is clear that the proposed method performs relatively better than the benchmark method for CHF to CNY exchange rate forecasting.

Figure 8 IMF component map of the CHF against the CNY exchange rate.

Figure 9 CHF to CNY exchange rate forecast and actual graph (for interpretation of the references to colour in this figure legend).

The EMD-BPNN model has a higher hit rate than other single reference models, while the prediction error is smaller. This shows that the EMD decomposition algorithm can improve the prediction accuracy of the neural network. This indicates that principal component analysis can reduce dimensionality and compress redundant data, improve prediction accuracy to a certain extent, and shorten the time of neural network training data.Notably, while predicting, one or more components of the highest frequency may be discarded. Therefore, the influence of high-frequency noise on prediction can be suppressed. Therefore, to eliminate the trend, except for the last component or components, all the extracted IMFs are added as decomposition results. Such a process can be easily combined with the smoothing of the results obtained if the highest frequency components have been discarded from the process of adding up the components.

Combination of deep learning and financial transaction

Deep learning can be used in various frequency trading, from low-frequency stock-picking models to high-frequency algorithmic trading models. Deep learning has been a thriving industry case at both levels of investment decision-making and transaction execution. For example, the hedge fund Cerebellum, which was established in 2009, manages assets of $90 billion, uses AI for adjunct forecasting and has been profitable every year since 2009. Man Group, one of the world’s most significant hedge funds, adopted AI to implement passive investment five years ago. Currently, the assets managed by AI have stable profits. Wall Street investment banks, such as Goldman Sachs and JPMorgan Chase, have also invested in AI stock-picking models. It is believed that machines can predict the results accurately through “deep learning” and reduce unnecessary transaction risks.

Deep learning is a method of learning the laws in massive data through DNN models. Deep learning ANNs are widely interconnected by numerous neurons, which are imitations of biological neural networks (brains). It is a nonlinear, distributed parallel processing, as well as a self-confirming algorithm model. Neurons are the fundamental units constituting a neural network. A neuron receives input signals sent from other neurons and produces inputs. In mathematics, a neuron is equivalent to a nonlinear transformation (excitation function). When a group of neurons is combined and has a hierarchical structure, a neural network model is formed. As deep learning develops, AI has made technical breakthroughs in many fields, such as image, speech, and natural speech processing. Currently, practical applications of AI are various and those in the financial field are also flourishing. In the meantime, deep learning is very suitable for financial prediction analysis in the context of big data. If deep learning is used, supplemented by a technique similar to knowledge maps, various events that have a significant influence on finance can be expressed in the form of knowledge maps. Then, features are automatically selected through deep networks for parameter and weight adjustments. The results can be more accurate and objective, and even those that have not been anticipated can be achieved.

Due to congregational psychology, humans are easily influenced by the surrounding environment during investments. The circular neural network has been widely applied in the field of natural language processing and has achieved great success. Such technologies make it possible to comprehend public opinion more accurately, thereby extracting the events that may affect the financial market. Combined with the above methods, various market states can be understood, providing users with better services.

Deep learning is used in areas such as financial risk control and big data credit. Hopefully, more new applications will emerge in the future. The use of deep learning in the financial field will lead to more intelligent management and consumption methods.

Empirical results and discussion

Here, an interval EMD algorithm is proposed based on the FEPA model. The research sample is the return rate of each index, and the forecast performance of the interval EMD model is tested empirically. The major conclusions include:

(1) The interval EMD model improves the prediction performance based on the FEPA model. Compared to the FEPA model, the prediction error of the interval EMD model is reduced. Compared to other reference models, the prediction error of the interval EMD model is much smaller. Compared to the FEPA model, the hit rate of the interval EMD model in predicting the closing market price increases by only 2%. However, the hit rate of the interval EMD model in predicting the highest and lowest prices increases by about 6% to 8% more than the FEPA model. Such increases show that the interval EMD model is useful in predicting FTS, especially the short-term fluctuation trends of the highest and lowest prices.

(2) Comprehensive and efficient utilization of transaction price information help improve forecast accuracy. In actual transactions, analysts will utilize comprehensive price information to predict the price trends of the future market. The empirical results suggest that if only the closing market price is considered, almost all data series are very similar to random walks. However, the interval EMD model that utilizes comprehensive transaction price information can improve the predictive ability of fluctuation trends in the stock index.

Conclusions

In this research, a prediction model is demonstrated for financial market forecasting. The FTSEMD can generate multi-layer IMF time series for FTS data. Then, the IMF series set is transformed by PCA, and its dimensionality is reduced to establish an ANN, which is used for prediction. The PSO algorithm is used to improve the prediction accuracy of the neural network model through parameter optimization. The algorithm approximates the global optimal by continually searching for current optimality. Moreover, the proposed model has the advantages of simple implementation, high precision, and fast convergence. The parameters are optimized effectively, and the priority of this model among other machine learning models is reported. In general, it is expected that the transaction process will not have much influence on the market. The trading delay should not be too long to lead the market price change toward an unfavourable direction. RMSE, MAPE, MAE, and TS are considered statistical indicators to demonstrate a fair comparative analysis to express the supremacy of the proposed forecasting model over RW.

Due to some objective limitations, only the data obtained by the Shanghai Stock Exchange Index, the Shenzhen Component Index, the Hansen Index, and the Dow Jones Industrial Average show regularity; nevertheless, the sample size is small to represent the entire market. Therefore, a more detailed investigation will be conducted in the future. Moreover, to overcome the limitation of EMD, we will use complete ensemble empirical mode decomposition with added noise (CEEMDAN) in our future work.

Supplemental Information

Supplemental Information 1 EMD code

Click here for additional data file.

Supplemental Information 2 Raw data for Figures 2, 4,6, and 8

Click here for additional data file.

Supplemental Information 3 Historical Data that used as EUR/CNY conversation rate

Click here for additional data file.

Supplemental Information 4 Historical Data that used as USD/CNY conversation rate

Click here for additional data file.

Supplemental Information 5 Historical Data that used as JPY/CNY conversation rate

Click here for additional data file.

Supplemental Information 6 Historical Data that used as CHF/CNY conversation rate

Click here for additional data file.

The author would like to thank the authors for the reference materials.

Additional Information and Declarations

Competing Interests

Author Contributions

Data Availability

Yixiao Jin is employed by Shanghai Yingcai Information Technology Ltd.

Zebin Jin conceived and designed the experiments, performed the experiments, analyzed the data, performed the computation work, prepared figures and/or tables, authored or reviewed drafts of the article, conceptualization, software, validation, formal analysis, and approved the final draft.

Yixiao Jin performed the computation work, prepared figures and/or tables, and approved the final draft.

Zhiyun Chen performed the computation work, prepared figures and/or tables, and approved the final draft.

The following information was supplied regarding data availability:

The raw data is available in the Supplemental Files.

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
