# Peer review of "Empirical mode decomposition using deep learning model for financial market forecasting"

_PeerJ Computer Science, doi:10.7717/peerj-cs.1076_

## Round 0.1 · original submission · Major Revisions

I have received reviews of your manuscript from three scholars who are experts on the cited topic. They find the topic very interesting; however, several concerns must be addressed regarding experimental results and design, and comparisons with currently existing methods. These issues require a major revision. Please refer to the reviewers’ comments listed at the end of this letter, and you will see that they are advising that you revise your manuscript. If you are prepared to undertake the work required, I would be pleased to reconsider my decision. Please submit a list of changes or a rebuttal against each point that is being raised when you submit your revised manuscript.

Thank you for considering PeerJ Computer Science for the publication of your research. We appreciate your submitting your manuscript to this journal.

Reviewer 1 ·

Basic reporting

This paper proposes a deep learning solution to the financial market forecasting.

Experimental design

There are few observations/suggestions to the authors.

1. Introduction section must end with the contributions of the proposed work.
2. Brief explanation on the dataset used is necessary.
3. As I understand this is a regression problem. Authors are suggested to include the error measures (Mean Absolute Error, Root Mean Square Error etc).
4. Good analysis is performed in the experimental section. The performance evaluations metrics must be applied if possible (namely accuracy and so on).
5. If possible, a comparative analysis of the proposed work vs existing work may be included.

Validity of the findings

This paper proposes a deep learning solution to the financial market forecasting. The article may be accepted for the possible publication after incorporating the suggestions mentioned.

Reviewer 2 ·

Basic reporting

The authors propose a method to forecast financial transactions by applying empirical mode decomposition to financial time series. Each intrinsic function is forecast with a back-propagation neural network. Then, all the predicted individual components are combined to obtain the overall predicted signal.

The use of EMD, instead of other signal decompositions, makes sense since financial time series are non-stationary and EMD can extract the main trend of these signals and suppress noise; however, EMD suffers from a major drawback, mixing mode. A better choice (suggestion for future work) would have been the use of other EMD versions that cope with such problem: CEEMDAN (complete ensemble empirical mode decomposition with added noise).

Experimental design

The number of days to forecast a future value depends on a sliding window of different sizes (10, 20, 30, 40, 50, 60, 70, 80, 90, 100). Are individual IMF forecasts made on actual input data within the sliding window or on previous predicted values within the window? Are IMF forecasts estimated in a single-day or multiple-day way? This part requires more explanations.

There is not information regarding the architecture of the neural network (number of layers and number of neurons per layer). How did the architecture was determined? These parts require more explanations.

Validity of the findings

No comment.

Additional comments

Questions arise regarding the architecture of the neural network, how hyper-parameters were chosen, and the process to forecast individual IMFs.

Reviewer 3 ·

Basic reporting

The paper studies the financial stock market forecasting problem and proposes the EMD based method.

There are some major problems as follows.

1. In the introduction, the long paragraphs describes the background of a wide range of financial market forecasting research, and the research is not focused enough. It should be more concise, focusing on the research work of stock forecasting.

2. The motivation for the proposed method is unclear. Why use the EMD based method?

3. The description of the method is unclear. In the method part, from the given Figure 1, the EMD method should be introduced first, then BNNN, and then PSO. The current structure should be revised.

Experimental design

The experimental results are not convincing enough. The experimental results are not compared with the current new method, and it is difficult to evaluate the effectiveness of the method.

Validity of the findings

No explicit findings are reported.

---

## Round 0.2 · accepted · Accept

I am pleased to inform you that your work has now been accepted for publication in PeerJ Computer Science.

Thank you for submitting your work to this journal.

Reviewer 1 ·

Basic reporting

Authors have incorporated the changes suggested. The paper may be accepted for the possible publication.

Experimental design

Authors have incorporated the changes suggested. The paper may be accepted for the possible publication.

Validity of the findings

NA

Additional comments

NA

Reviewer 2 ·

Basic reporting

The topic is interesting and the research presented is of adequate quality. I had concerns that were addressed by the authors.

Experimental design

The authors have given due attention to previous concerns.

Validity of the findings

The authors have given due attention to previous concerns.

Additional comments

After the prediction of each component, these are added to obtain the final forecast of the original signal. The current study is interesting. In general, the main conclusions are supported by the figures and text.